# Persistent inflammation during anti-tuberculosis treatment with diabetes comorbidity

Nathella Pavan Kumar[1†], Kiyoshi F Fukutani[2,3,4†], Basavaradhya S Shruthi[5], Thabata Alves[2,6], Paulo S Silveira-Mattos[2,4], Michael S Rocha[2], Kim West[7], Mohan Natarajan[8], Vijay Viswanathan[5], Subash Babu[1], Bruno B Andrade[2,3,4,6‡], Hardy Kornfeld[7‡*]

[1]National Institutes of Health, National Institute for Research in Tuberculosis, International Center for Excellence in Research, Chennai, India; [2]Multinational Organization Network Sponsoring Translational and Epidemiological Research (MONSTER), Fundação José Silveira, Salvador, Brazil; [3]Instituto Gonçalo Moniz, Fundação Oswaldo Cruz, Salvador, Brazil; [4]Faculdade de Tecnologia e Ciências, Salvador, Brazil; [5]Prof. M. Viswanathan Diabetes Research Center, Chennai, India; [6]Universidade Salvador, Laureate Universities, Salvador, Brazil; [7]University of Massachusetts Medical School, Worcester, United States; [8]National Institute for Research in Tuberculosis, Chennai, India

**Abstract** Diabetes mellitus (DM) increases risk for pulmonary tuberculosis (TB) and adverse treatment outcomes. Systemic hyper-inflammation is characteristic in people with TB and concurrent DM (TBDM) at baseline, but the impact of TB treatment on this pattern has not been determined. We measured 17 plasma cytokines and growth factors in longitudinal cohorts of Indian and Brazilian pulmonary TB patients with or without DM. Principal component analysis revealed virtually complete separation of TBDM from TB individuals in both cohorts at baseline, with hyper-inflammation in TBDM that continued through treatment completion at six months. By one year after treatment completion, there was substantial convergence of mediator levels between groups within the India cohort. Non-resolving systemic inflammation in TBDM comorbidity could reflect delayed lesion sterilization or non-resolving sterile inflammation. Either mechanism portends unfavorable long-term outcomes including risk for recurrent TB and for damaging immune pathology.
DOI: https://doi.org/10.7554/eLife.46477.001

*For correspondence:
Hardy.Kornfeld@umassmed.edu

†These authors contributed equally to this work
‡These authors also contributed equally to this work

Competing interests: The authors declare that no competing interests exist.

## Introduction

It is now well established that diabetes mellitus (DM) is associated with increased the risk to become infected with *Mycobacterium tuberculosis* (*Mtb*), to progress from latent infection to active pulmonary tuberculosis (TB) disease, and to suffer adverse TB outcomes including delayed sputum conversion on treatment, treatment failure, death, and recurrent TB (*Critchley et al., 2017*). The biochemical and cellular mechanisms of increased TB susceptibility in DM are incompletely understood. Emerging evidence ties the complication of diabetic immunopathy to the well-studied diabetic complications of microvascular, macrovascular and renal disease driven by pathways primary driven by chronic hyperglycemia and related oxidative stress (*Giacco and Brownlee, 2010*; *Martinez and Kornfeld, 2014*).

**eLife digest** Tuberculosis is the leading cause of death from infection worldwide. The bacteria that causes tuberculosis infect one in every four people on the planet, though most never develop the disease. People with diabetes are more likely to develop tuberculosis and they develop more severe symptoms, which may contribute to further spread of the disease. As diabetes rates are growing worldwide, particularly in countries with a high burden of tuberculosis, it is becoming to increasingly important to understand how these two conditions interact.

People with diabetes often have more severe inflammation at the time they are diagnosed with tuberculosis than tuberculosis patients without diabetes. Inflammation can cause permanent lung damage in patients with tuberculosis, which can have serious consequences. Learning more about how treatment for tuberculosis affects inflammation in people with diabetes could help improve the outcomes for these patients.

Now, Kumar, Fukutani et al. show that people with diabetes experience higher levels of inflammation than patients without diabetes throughout the course of treatment for tuberculosis. The analysis compared 17 markers of inflammation in tuberculosis patients with and without diabetes at diagnosis, and at two time periods during the 6-month course of treatment. Kumar, Fukutani et al. also looked at two separate groups of patients, one from India and one from Brazil. Inflammation was measured in the patients in India one year after the completion of treatment. One-year after treatment for tuberculosis in India, inflammation levels were the same in patients with and without diabetes.

Persistently higher levels of inflammation likely explain why patients with diabetes experience more severe symptoms and suggests they may have more permanent lung damage after tuberculosis. Scientists are currently developing new treatments that can be used with antibiotics to more quickly cure tuberculosis and protect the lungs by reducing inflammation. Patients with diabetes and tuberculosis may benefit from these new treatments or from existing drugs like metformin or statins that may reduce inflammation.

DOI: https://doi.org/10.7554/eLife.46477.002

Pulmonary TB disease with concurrent DM is associated with a higher burden of immune pathology and systemic inflammation compared to TB in euglycemic hosts (*Martinez and Kornfeld, 2014*). Restrepo et al. were the first to describe increased expression of a broad range of cytokines from antigen-stimulated whole blood from patients with concurrent TB and DM as compared to TB alone

**Table 1.** Clinical and demographic characteristics of participants in the India cohort.

| Characteristic | TBDM | TB | P value |
|---|---|---|---|
| | n = 43 | n = 44 | |
| Age, median years (IQR) | 46 (38–52) | 39.5 (30–47) | 0.0146 |
| Male sex, no. (%) | 32 (74.4) | 37 (84) | 0.0408 |
| Smoking status, no. (%) | | | 0.0584 |
| Current smoker | 5 (11.6) | 12 (27.3) | |
| Former smoker | 9 (21) | 13 (29.5) | |
| Never smoked | 29 (67.4) | 19 (43.2) | |
| Alcohol use, no. (%) | | | 0.7410 |
| Current use | 11 (25.6) | 21 (47.7) | |
| Former user | 16 (37.2) | 10 (22.7) | |
| Never used | 16 (37.2) | 13 (29.6) | |
| BMI kg/m$^2$, median (IQR) | 20.3 (163–23) | 16·4 (15.2–18.3) | <0.0001 |
| HbA1c %, median (IQR) | 10.0 (7.3–11.8) | 5.6 (5.4–5.8) | <0.0001 |
| Vitamin D ng/dL, median (IQR) | 15 (9.3–24) | 17 (13–27) | 0.1502 |

DOI: https://doi.org/10.7554/eLife.46477.003

(*Restrepo et al., 2008*). This was subsequently verified by several studies reporting higher plasma cytokine levels and higher frequencies of cytokine-expressing T cells assessed by flow cytometry in samples from individuals with TBDM comorbidity (*Kumar et al., 2013a*; *Kumar et al., 2013b*; *Prada-Medina et al., 2017*). Published studies have focused on samples obtained at baseline, before or shortly after the initiation of anti-TB treatment (ATT). We hypothesized that the hyperinflammatory phenotype of TBDM comorbidity is driven by a higher lung bacterial burden and/or by impaired counter regulatory mechanisms that normally limit bystander tissue injury from protective immune responses. To test the prediction that the DM would be associated with delayed resolution of inflammation during TB treatment, we conducted a longitudinal assessment of plasma cytokine and growth factor levels in cohorts of adult pulmonary TB patients with and without DM recruited in India and Brazil.

## Results

### Characteristics of the study populations

Adults suspected to have pulmonary TB disease were screened for participation in the Effects of Diabetes on Tuberculosis Severity (EDOTS) study on presentation to the government clinic system in Chennai, India. Candidates conforming to the inclusion and exclusion criteria where tested for glycemic status based on medical history, oral glucose tolerance test (only for those without prior DM history), and glycohemogloblin (HbA1c; for all enrolled participants). Follow-up visits occurred at monthly intervals during the 6 month course of ATT for drug-sensitive TB and then quarterly through the final study visit one year after ATT completion (month-18). Enrolled participants were withdrawn from the study and not included in the current analysis if their baseline sputum culture was negative or if it was positive for multi-drug resistant *Mtb*.

Selected demographic, anthropometric and behavioral variables of the India cohort are shown in *Table 1*. Compared to the normoglycemic TB group, the TBDM group had higher median age and higher body mass index (BMI). The median and interquartile range of BMI in the normoglycemic TB group fell below the WHO cutoff for undernutrition of 18.5 kg/m$^2$ (*WHO Expert Consultation,*

**Table 2.** Characteristics of India cohort participants with KDM vs NDM at enrollment.

| Characteristic | KDM | NDM | P value |
|---|---|---|---|
| Male, n (%) | 20 (71.4) | 12 (42.9) | 0.7190 |
| Age, Median (IQR) | 45.5 (38.0–52.0) | 48.0 (38.0–48.0) | 0.5072 |
| BMI, Median (IQR) | 21.7 (18.9–23.6) | 18.9 (16.1–20.8) | 0.0079 |
| Smoking history, n (%) | | | 0.7354 |
| Yes | 10 (35.7) | 4 (14.3) | |
| No | 18 (64.3) | 11 (39.3) | |
| Current drinker, n (%) | | | 0.9999 |
| Yes | 7 (25.0) | 4 (14.3) | |
| No | 21 (75.0) | 11 (39.3) | |
| Cavitation, n (%) | | | 0.133 |
| Yes | 5 (17.9) | 6 (21.4) | |
| No | 24 (85.7) | 8 (28.6) | |
| Bilateral lung lesion, n (%) | | | 0.3319 |
| Yes | 16 (57.1) | 5 (17.9) | |
| No | 13 (46.4) | 9 (32.1) | |

KDM, known DM prior to enrollment; NDM, newly diagnosed DM at enrollment screening.

Data were compared using the chi-squared test except for age, which was compared using the Mann-Whitney *U* test.

DOI: https://doi.org/10.7554/eLife.46477.004

Table 3. Clinical and demographic characteristics of participants in the Brazil cohort.

| Characteristic | TBDM | TB | P value |
|---|---|---|---|
| | n = 25 | n = 26 | |
| Age, median years (IQR) | 45 (30.5–49.5) | 46 (37–56) | 0.131 |
| Male sex, no. (%) | 13 (52) | 13 (50) | >0.999 |
| Smoking status, no. (%) | | | 0.162 |
| Current smoker | 10 (40) | 8 (30.7) | |
| Former smoker | 3 (12) | 9 (34.6) | |
| Never smoked | 12 (48) | 9 (34.6) | |
| Alcoholism, no. (%) | 10 (40) | 13 (50) | 0.473 |
| BMI kg/m$^2$, median (IQR) | 19.5 (18.3–49.5) | 20.2 (18.7–22.6) | 0.114 |
| HbA1c %, median (IQR) | 8.8 (7.3–10.2) | 5.2 (4.7–5.5) | <0.0001 |

Alcoholism defined by CAGE questionnaire.
DOI: https://doi.org/10.7554/eLife.46477.005

2004). Trends for higher likelihood of self-reported current or past smoking in the normoglycemic TB group did not reach statistical significance. There was no difference in the proportion of TB or TBDM participants who reported current or past alcohol consumption. The median HbA1c in TBDM participants was 10.0%, indicating poor glycemic control. There were no statistically significant differences in sex, age, smoking or alcohol consumption between TBDM participants with DM newly diagnosed during screening for this (NDM) study vs known DM diagnosis prior to incident TB (KDM), but median BMI was higher in the KDM group (*Table 2*). All participants in the non-diabetic TB group were classified as euglycemic based on oral glucose tolerance test but sixteen had HbA1c $\geq$ 5.8%, placing them in the prediabetic range by American Diabetes Association criteria. There was no difference in median 25-hydroxyvitamin D levels between TB and TBDM participants, although the median for both groups fell below a common threshold for insufficiency (20 ng/mL) (*Thacher and Clarke, 2011*). There were no significant differences between groups in the TB radiographic severity score, the presence of lower lung zone lesions or cavitary lesions.

The Brazil cohort differed from the India cohort in several respects (*Table 3*). In that cohort, DM status was classified based only on HbA1c, there was no difference in median age between TBDM and TB participants, roughly equal proportions of male and female participants, and no differences between groups in smoking or alcohol consumption. There was no difference in median BMI between TBDM and TB participants in the Brazil cohort and nearly all participants had BMI above the undernutrition cutoff of 18.5 kg/m$^2$. Median HbA1c was lower in the Brazilian TBDM and TB groups compared to their Indian counterparts, but glycemic control was nonetheless poor ($\geq$7.5%) in the majority of Brazilian TBDM participants.

A comparison of demographic and clinical characteristics between the cohorts from Brazil and India is shown in *Table 4* and further stratified into TB and TBDM subgroups within each cohort in *Table 5*. Statistically significant differences included a higher proportion of female sex and higher BMI in the Brazil cohort, while the cohorts did not differ with respect to alcohol or consumption or smoking. Sputum acid-fast bacilli smear grades were significantly higher in TBDM than TB participants from India with a similar trend in the Brazil cohort that did not reach statistical significance (*Figure 1a*). Sputum smear grade differences between TBDM and TB were highly significant when the combined participants from both countries were considered (*Figure 1b*).

## TBDM and TB are associated with distinct plasma cytokine profiles

Plasma levels of seventeen cytokines and growth factors were measured in samples from participants in the Indian TBDM and TB groups at baseline (pre-ATT), after the intensive phase of ATT (month-2), at the completion of ATT (month-6), and one year after TB treatment completion (month-18). Hierarchical cluster analysis of log2-transformed and z-score normalized values for each analyte identified two main clusters exclusively comprised of TBDM or TB, respectively, at baseline (*Figure 2a*). The TBDM group exhibited higher levels of nearly all analytes compared to the TB group. The two

**Table 4.** Comparison of Brazil and India cohort characteristics.

|  | Brazil | India | P value |
|---|---|---|---|
| Age, Median (IQR) | 46.0 (34.0–50.0) | 43.0 (32.0–54.0) | 0.5400 |
| Male, n (%) | 26 (51.0) | 69 (79.3) | 0.0011 |
| BMI, Median (IQR) | 17.8 (15.7–21.0) | 19.8 (18.6–22.2) | 0.0001 |
| Smoking history, n (%) |  |  | 0.2871 |
| Yes | 18 (35.3) | 17 (26.2) |  |
| No | 33 (64.7) | 48 (73.9) |  |
| Alcohol, n (%) |  |  | 0.1409 |
| Yes | 23 (45.1) | 32 (36.8) |  |
| No | 23 (45.1) | 55 (63.2) |  |
| Lung lesions, n (%) |  |  | 0.6336 |
| Unilateral | 26 (51.0) | 48 (55.2) |  |
| Bilateral | 25 (49.0) | 39 (44.8) |  |
| Cavitation, n (%) |  |  | 0.2718 |
| Yes | 33 (64.7) | 64 (73.6) |  |
| No | 18 (35.3) | 23 (26.4) |  |
| AFB smear grade, n (%) |  |  | <0.0001 |
| 0 | 9 (17.6) | 0 (0) |  |
| 1+ | 13 (25.4) | 43 (52.4) |  |
| 2+ | 14 (27.4) | 35 (42.7) |  |
| ≥3 + | 15 (29.4) | 4 (4.9) |  |

Data were compared using the chi-squared test except for age and BMI, which were compared using the Mann-Whitney $U$ test.

DOI: https://doi.org/10.7554/eLife.46477.008

discrete clusters comprised only of TBDM or TB individuals were maintained at the month-2 and month-6 timepoints but were lost by month-18. The separation between TBDM and TB groups based on plasma cytokine levels was also evident on principal component analysis (PCA; *Figure 2b*). By PCA, there was nearly complete separation of groups at baseline, with a slight trend towards overlap at month-2 and month-6, culminating in near complete merger of values for the two groups at month-18, albeit with some outliers among TBDM participants.

An identical panel of plasma analytes was measured in samples from the Brazil cohort collected at pre-ATT baseline, at treatment month-2, and at treatment completion (month-6). Results for the Brazilian participants were remarkably like those obtained in the India cohort, with complete separation of the TBDM and TB groups at baseline and only partial overlap at the later time points during treatment (*Figure 1c and d*). These results show that the unique pattern of systemic hyperinflammation associated with concurrent TB and DM was present in Indian and Brazilian populations, despite considerable differences in host genetic, demographic, and behavioral characteristics, and in the local prevalent *M. tuberculosis* strains in the two study populations (*Suzana et al., 2017*; *Vasconcellos et al., 2014*).

## Longitudinal trends in plasma cytokine levels within and between groups

To compare the temporal pattern of plasma cytokine levels within the TBDM or TB groups over the course of ATT, log2 transformed data were analyzed using one-way ANOVA with linear trend post-test. Results that remained statistically significant (FDR 1%) were included the heatmaps for the India and Brazil cohorts. The Indian TBDM group exhibited a linear trend of deceasing plasma levels of most pro-inflammatory cytokines during and after ATT, with rising levels of IL-5, IL-10, IL-13, and TGF-b (*Figure 3a*). Similar trends were identified in the Brazilian TBDM group (*Figure 4a*). The

**Table 5.** Characteristics of normoglycemic TB participants and TBDM participants in the India and Brazil cohorts.

| Characteristics | TB | | | TBDM | | |
|---|---|---|---|---|---|---|
| | India | Brazil | P value | India | Brazil | P value |
| Age, Median (IQR) | 39.5 (30.0–47.2) | 45.0 (30.5–49.5) | 0.6390 | 47.0 (38.0–51.0) | 46.0 (37.0–56.0) | 0.5565 |
| Male, n(%) | 32 (72.7%) | 13 (50%) | 0.3001 | 37 (84,09%) | 13 (52%) | 0.0058 |
| BMI, Median (IQR) | 16.4 (15.3–18.3) | 19.5 (18.3–20.6) | 0.2579 | 20.32 (16.6–23.0) | 20.20 (18.7–22.6) | <0.0001 |
| Smoking, n (%) | | | 0.6148 | | | 0.1309 |
| Yes | 25 (56.8%) | 17 (65.4%) | | 14 (32.6%) | 13 (52%) | |
| No | 19 (43.2%) | 9 (346%) | | 29 (67.4%) | 12 (48%) | |
| Alcohol, n (%) | | | >0.9999 | | | 0.2786 |
| Yes | 21 (47.7%) | 13 (50%) | | 11 (25.6%) | 10 (40%) | |
| No | 23 (52.3%) | 13 (50%) | | 32 (74.4%) | 15 (60%) | |
| Lung lesion, n (%) | | | >0.9999 | | | 0.4535 |
| Unilateral | 26 (591%) | 16 (61.5%) | | 22 (51,16%) | 10 (40%) | |
| Bilateral | 18 (40.9%) | 10 (38.5%) | | 21 (48,84%) | 15 (60%) | |
| Cavitation, n (%) | | | 0.0212 | | | 0.0003 |
| Yes | 12 (27.3%) | 15 (57.7%) | | 11 (25.6%) | 18 (72%) | |
| No | 32 (72.7%) | 11 (42.3%) | | 32 (74.4%) | 7 (28%) | |
| AFB smear, n (%) | | | 0.0024 | | | <0.0001 |
| 0 | 0 (0%) | 6 (23.1%) | | 0 (0%) | 3 (12%) | |
| 1+ | 26 (61.9%) | 9 (34.6%) | | 17 (42,5%) | 4 (16%) | |
| 2+ | 13 (30.9%) | 6 (23.1%) | | 22 (55%) | 8 (32%) | |
| ≥3 + | 3 (7.1%) | 5 (19.2%) | | 1 (2,5%) | 10 (40%) | |

Data were compared using the chi-square test except foe age as BMI, which were compared using the Mann-Whitney *U* test.

DOI: https://doi.org/10.7554/eLife.46477.009

trends within the normoglycemic TB groups from India (*Figure 3a*) and Brazil (*Figure 4a*) where mutually consistent and differed from the TBDM groups most notably with declining levels of IL-10.

Analysis of plasma cytokines accounting for differences between the TBDM and TB groups of the India cohort is shown in *Figure 3b*. There was a statistically significant difference in the levels of

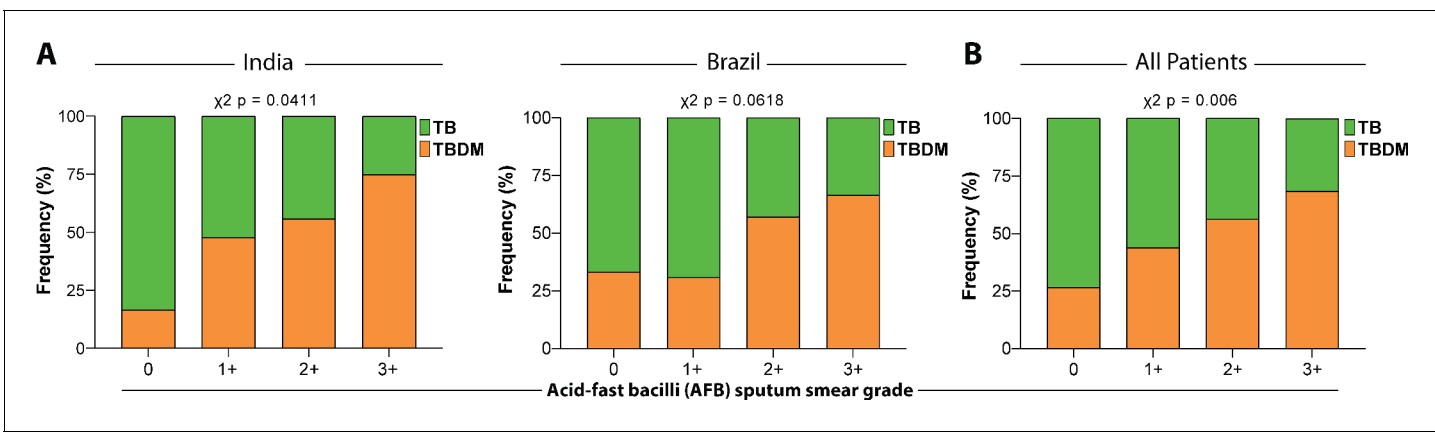

**Figure 1.** Mycobacterial burden in sputum smear stratified by glycemic status. Sputum AFB smear grade in participants with DM comorbidity (TBDM, *Orange*) or euglycemia (TB, *Green*). (**A**) Frequency of India cohort participants (left panel) and Brazil cohort participants (right panel) with different AFB smear grades ranging from 0 to ≥3 + . (**B**) Frequency of TB and TBDM participants from the combined Indian and Brazil cohorts with different AFB smear grades. Data were analyzed using Pearson's chi-squared test.

DOI: https://doi.org/10.7554/eLife.46477.006

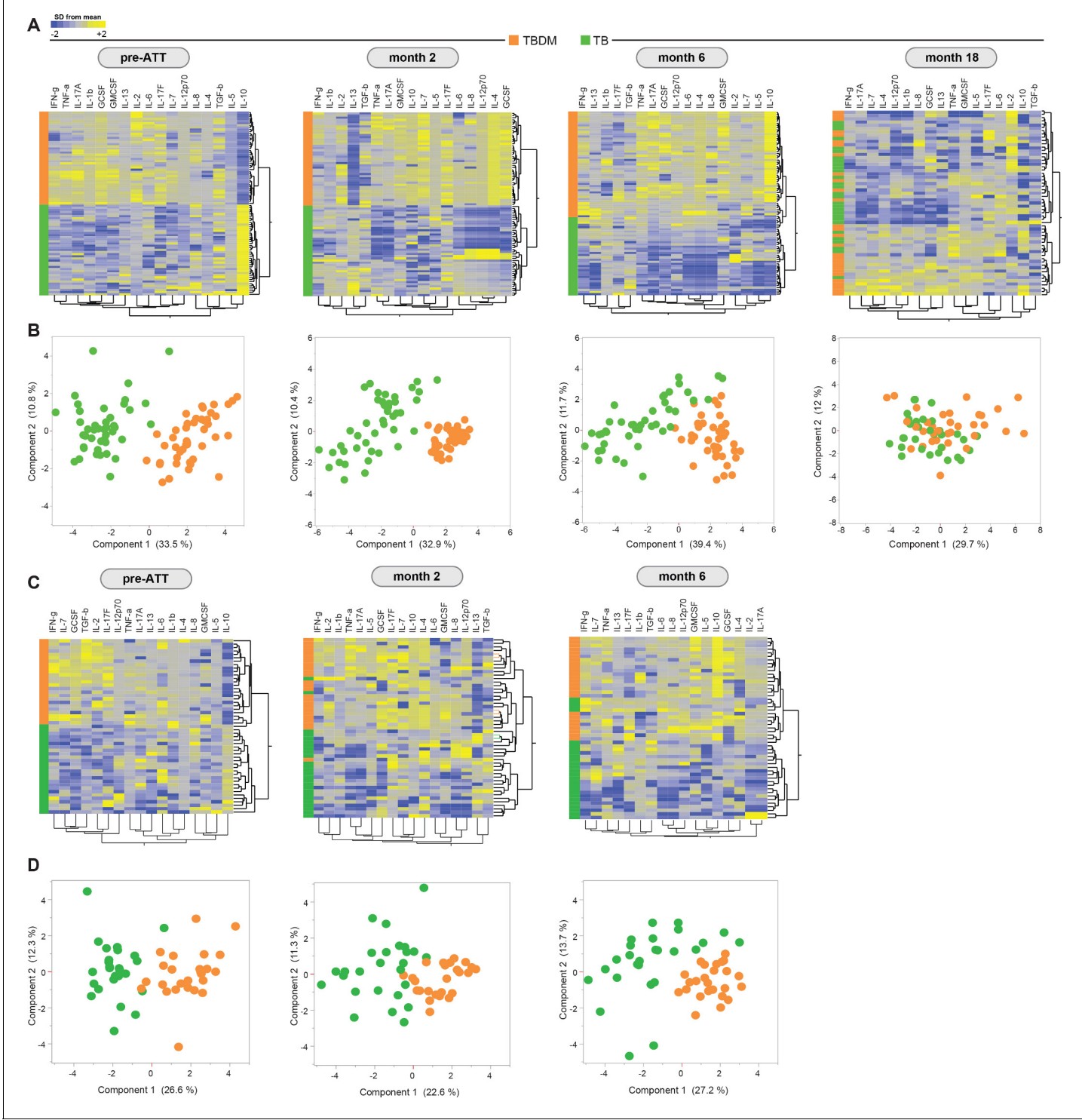

**Figure 2.** Prospective assessment of plasma biomarkers in pulmonary TB patients with or without concurrent diabetes undergoing anti-TB treatment. Hierarchical cluster analysis (Ward's method with 100x bootstrap) of z-score normalized, log-transformed values for each plasma analyte from the Indian and Brazil cohorts (A and C, respectively) at the indicated timepoints of antimicrobial treatment for drug-sensitive pulmonary TB. In the heatmaps, yellow color represents the highest values whereas blue color indicates the lowest values measured for each analyte. Principal component analysis was performed to show the distribution of data from the India cohort and Brazil cohorts (B and D, respectively) on simultaneous assessment of variables shown in the heatmaps.

DOI: https://doi.org/10.7554/eLife.46477.010

fourteen out of seventeen cytokines measured in the India cohort at baseline, of which only one (IL-10) was lower in TBDM than TB. Notably, the fold-difference in baseline IL-10 levels exceeded that of all other analytes at all timepoints. The pattern of higher cytokine levels in TBDM than TB individuals was mostly maintained at month-2 except that IL-10 changed from being significantly lower in TBDM than TB at baseline to significantly higher as compared to TB, while IL-13 that changed from being slightly higher in TBDM at baseline to markedly lower than TB at month-2. The direction and magnitude of differences in levels of most other measured cytokines were similar in month-6 compared to month-2, but at month-18 only six cytokines were present at significantly higher levels in TBDM and the magnitude of fold-differences was much reduced. There were no significant differences in cytokine levels between TBDM participants with DM newly diagnosed during screening as compared those known to have DM prior to enrollment (*Figure 3—figure supplement 1*).

Analysis of between-group cytokine levels in the Brazil cohort showed only five of seventeen total analytes present at significantly different levels in TBDM vs TB participants (*Figure 4b*). Like the India cohort, Brazilian TBDM participants had significantly lower levels of IL-10 at baseline. The levels of IFN-g, IL-17F, GCSF, and TGF-b were significantly higher in Brazilian TBDM than TB participants at baseline. By month-2, IL-10 reversed from significantly higher to significantly lower in the TBDM group, as was seen in the India cohort. At the ATT completion (month-6), eight analytes were present at significantly different levels in Brazilian TBDM vs TB participants. Of these, seven were higher

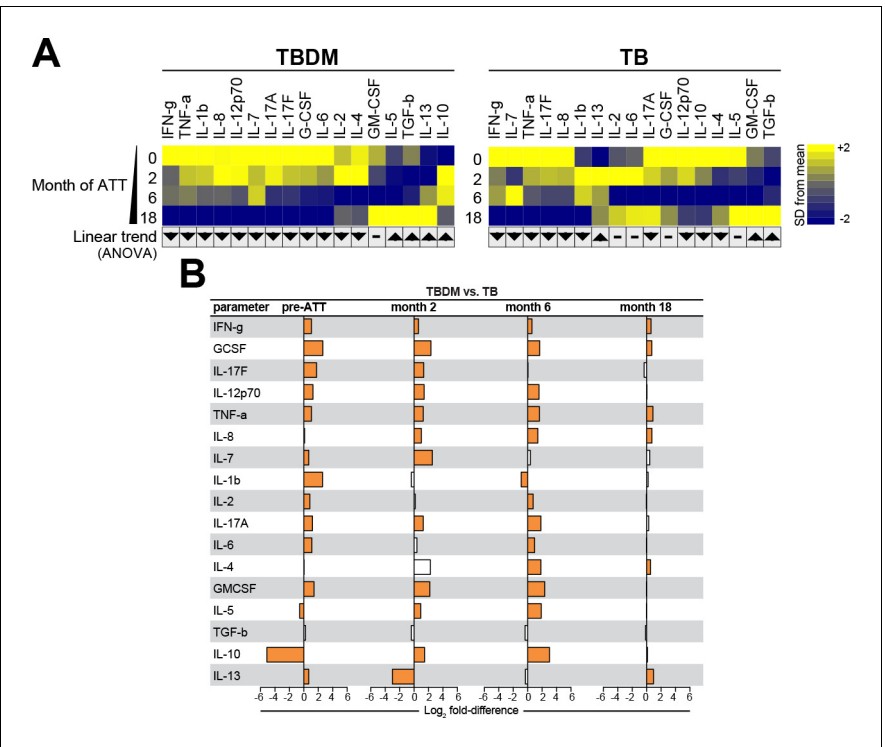

**Figure 3.** Biomarker profiles during anti-TB treatment in TBDM comorbidity from the India cohort. (**A**) Mean log-transformed values for the indicated analytes were calculated and z-score normalized. Heatmaps with values from the India cohort grouped using hierarchical clustering (Ward's method with 100x bootstrap) was used to illustrate the overall variation in plasma concentrations over time. In addition, one-way ANOVA with linear trans ad hoc test was used to test the direction of variation in each analyte's concentration between study timepoints. Direction of the arrows highlight statistically significant trends, while "- "denotes differences with P values $\geq$ 0.05. (**B**) Mean fold-difference of analyte levels between the TBDM and TB groups in the India cohort. Orange bars indicated statistically significant differences (p<0.05).
DOI: https://doi.org/10.7554/eLife.46477.012

The following figure supplement is available for figure 3:

**Figure supplement 1.** Biomarker profiles in India cohort TBDM participants with KDM vs NDM at enrollment.
DOI: https://doi.org/10.7554/eLife.46477.013

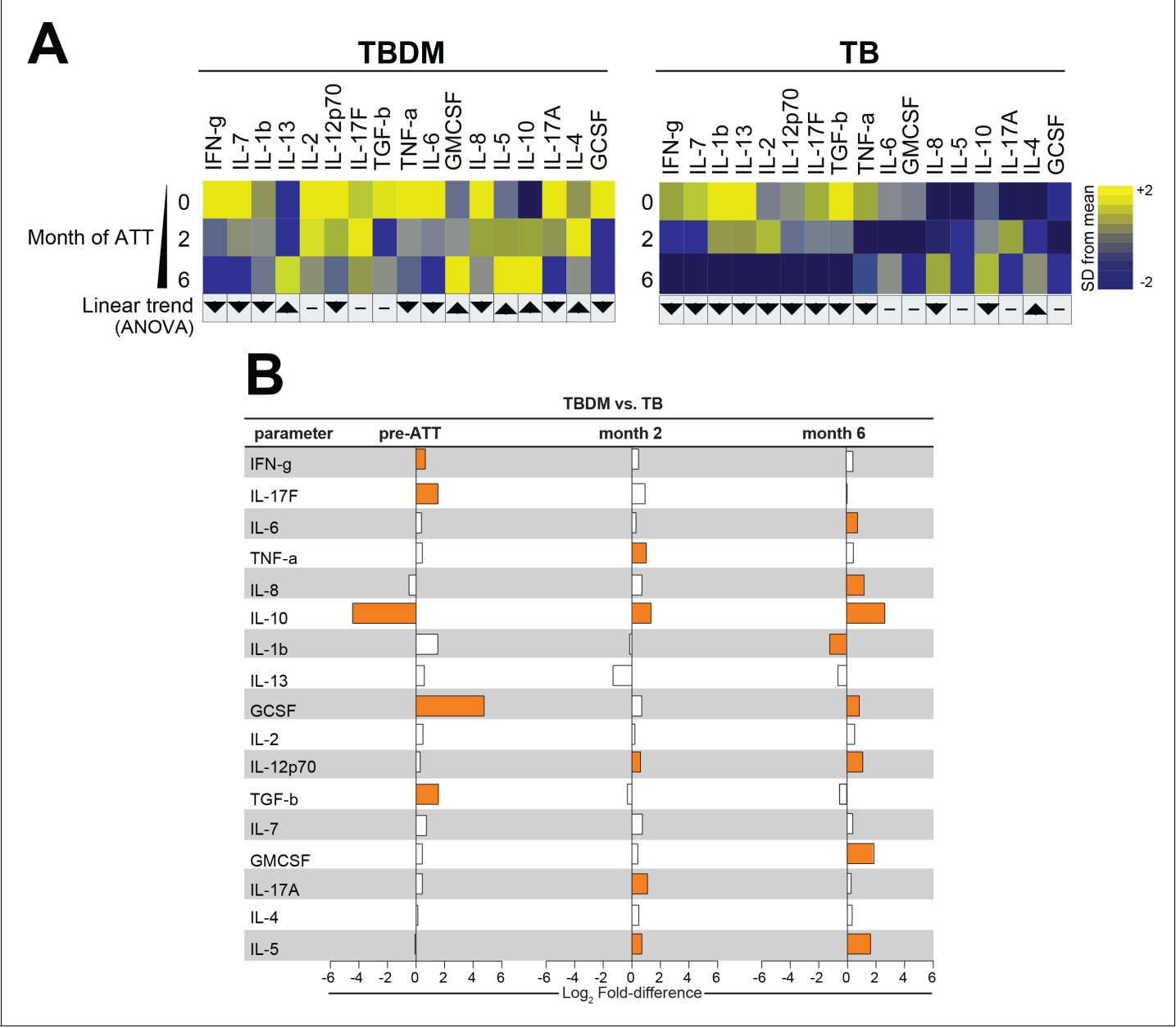

**Figure 4.** Biomarker profiles during anti-TB treatment in TBDM comorbidity from the Brazil cohort. (**A**) Mean log-transformed values for the indicated analytes were calculated and z-score normalized. Heatmaps with values from the Brazil cohort grouped using hierarchical clustering (Ward's method with 100x bootstrap) was used to illustrate the overall variation in plasma concentrations over time. In addition, one-way ANOVA with linear trans ad hoc test was used to test the direction of variation in each analyte's concentration between study timepoints. Direction of the arrows highlight statistically significant trends, while "- "denotes differences with P values $\geq$ 0.05. (**B**) Mean fold-difference of analyte levels between the TBDM and TB groups in the Brazil cohort. Orange bars indicated statistically significant differences (p<0.05).
DOI: https://doi.org/10.7554/eLife.46477.014

in TBDM than TB, with only IL-1b lower in TBDM. Taken together, these analyses indicated that TBDM individuals had higher levels of circulating pro-inflammatory mediators compared to TB individuals at baseline, and this systemic hyper-inflammatory state persisted through the course of TB treatment with a temporal shift in relative IL-10 levels.

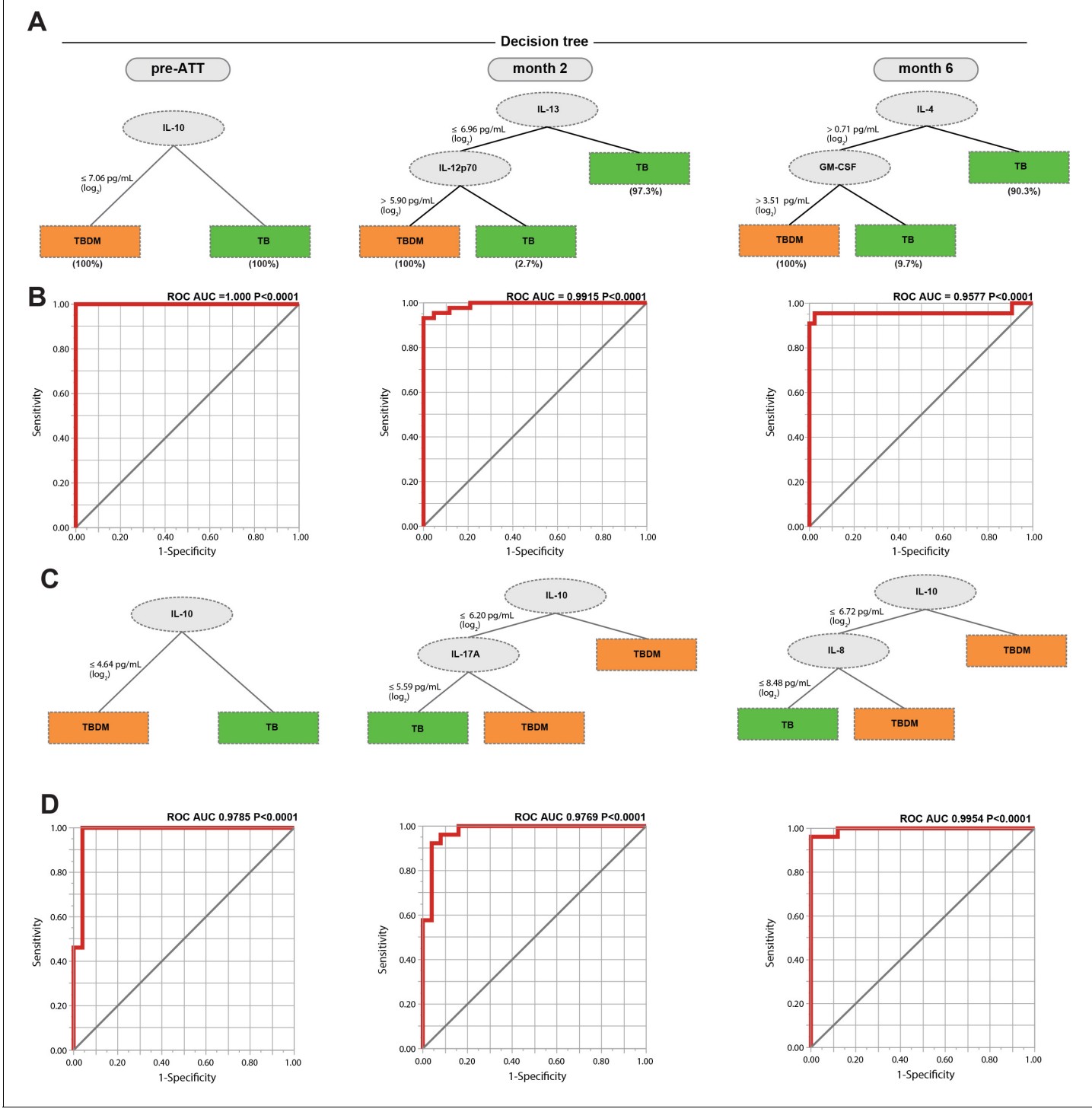

**Figure 5.** Identification of biomarkers showing the strongest associations with TBDM comorbidity. Decision tree analysis shows the analytes (or combination) that exhibited the highest accuracy in discriminating TBDM from TB in the India and Brazil cohorts (**A and C**, respectively). Receiver operator characteristics curves were employed to quantify the accuracy of single or combined biomarkers int the India and Brazil cohorts (**B and D**, respectively).

DOI: https://doi.org/10.7554/eLife.46477.015

## Biomarkers discriminating TBDM from TB over the course of TB treatment

Decision tree analysis of plasma cytokine data was performed to identify those biomarkers that differentiated TBDM from TB individuals with the highest accuracy and minimal numbers of markers, which might reflect mechanistic differences between groups within the cohorts (*Figure 5*). At baseline, IL-10 exhibited the strongest discriminating effect, with lower levels in the TBDM than the TB groups in both the India and Brazil cohorts (*Figure 5a and c*, respectively). The patterns in the Indian and Brazilian participants diverged at later time points. At month-2, lower IL-13 along with higher IL-12p70 levels powered the difference between TBDM and TB in the India cohort. The pattern shifted again at month-6 with higher levels of IL-4 and GM-CSF associating with TBDM. In contrast, IL-10 remained the strongest discriminating factor in the Brazil cohort at all time points evaluated, along with lower IL-17A in the TB group at month-2 and lower IL-8 in the TB group month-6.

Receiver operating characteristic (ROC) analysis showed the single or combined cytokines that accurately defined the difference between TBDM and TB in both cohorts at the three timepoints, with area under the curves $\geq$ 0.9577 and P values < 0.0001 (*Figure 5b and d*). The underlying biological mechanisms can only be inferred from the available data, but low baseline levels of the anti-inflammatory cytokine IL-10 in TBDM comports with higher baseline levels of several pro-inflammatory cytokines in that group (*Sugimoto et al., 2016*). Overall, the dynamic pattern demonstrated by these results is consistent with prolonged inflammation and delayed relative rise in pro-resolving cytokines (IL-10, IL-4, IL-13) (*Ortega-Gómez et al., 2013*) in the TBDM group.

## Association of plasma cytokines with HbA1c and radiographic severity of TB

A validated method for grading TB severity by chest x-ray interpretation was used to score images obtained at baseline, month-6, and month-18 in the India cohort of TBDM and TB individuals (*Ralph et al., 2010*). There was considerable individual variation at all timepoints but no statistically significant difference in median radiographic scores between groups at any timepoint (*Figure 6a*). Spearman correlation matrices were built to examine associations between the radiographic score value at each time point, or the difference in score value between the indicated time points (fold variation), and the individual plasma analyte levels in the entire cohort. The association of baseline HbA1c with plasma analytes at each time point was also examined. As shown in *Figure 6b*, the strongest associations identified were between plasma biomarkers at baseline with HbA1c, particularly TNF-α, IL-2, IL-17F, and IL-1β (positive correlations) and IL-5 and IL-10 (negative correlations). Biomarkers measured at month-6 remained strongly associated with the HbA1c measured at baseline, but with shift from strong positive to weak negative correlation with IL-1 and from strong negative to moderate positive with IL-10. While some statistically significant correlations of radiographic scores or change in radiographic score were identified, particularly with month-18 biomarker levels, these were mostly moderate to weak correlations and did not demonstrate a consistent pattern or trend. The number of analytes having significant correlation with baseline HbA1c by this analysis changed from sixteen at baseline, to ten at month-6, and one only one (positive correlation with IL-7) at month-8. Interestingly, IL-7 is known to maintain strong cellular immune responses for months, in contrast to more short-lived responses to IL-2 (*Lynch and Miller, 1994*).

## Discussion

Increased circulating levels of numerous cytokines and growth factors, many linked to protective immunity, typifies TBDM comorbidity at the time of TB diagnosis (*Kumar et al., 2013a*; *Prada-Medina et al., 2017*; *Restrepo et al., 2008*). Our goal for the current study was to investigate longitudinal trends in plasma levels of these mediators over the course of treatment for drug-sensitive pulmonary TB in participants rigorously classified as diabetic or normoglycemic at baseline. Plasma samples from two ongoing observational cohort studies, one in India and one in Brazil, were available for this investigation. Seventeen plasma mediators were measured by a combination of Luminex and conventional ELISA. Results from baseline samples were consistent with the previously reported hyper-inflammatory state of TB with concurrent DM prior to the initiation of ATT. Despite clinically efficacious antimicrobial chemotherapy, we found that the hyperinflammatory profile of TBDM

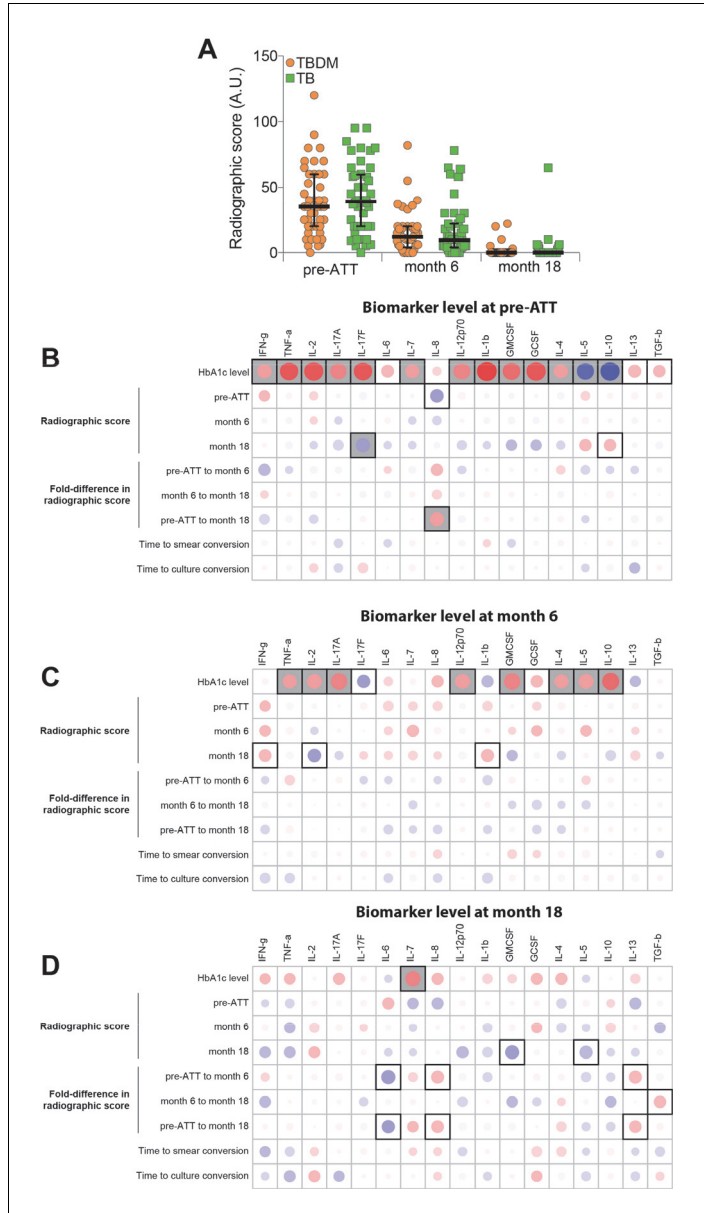

**Figure 6.** Associations between radiographic scores, HbA1c, and systemic inflammatory profiles in the India cohort. (**A**) Change in radiographic score values before and at the indicated timepoints after the initiation of antimicrobial treatment for pulmonary TB. Dots represent individual participant values and horizontal lines indicate median values. Values were compared between TBDM and normoglycemic TB groups using the Mann-Whitney *U* test. (**B–D**). Spearman correlation matrices were built to examine associations between absolute radiographic score values or difference in score values between the indicated timepoints (fold-variation) and the indicated plasma analyte value at each study timepoint. The Spearman rank values are shown in a heatmap scale. Statistically significant correlations (p<0.05) are highlighted in bold squares. Gray squares indicate significant correlations after adjusting for false discovery rate (FDR 1%). Red dots represent positive correlations while blue dots represent negative correlations.

DOI: https://doi.org/10.7554/eLife.46477.016

participants persisted in both Indian and Brazil cohorts through the intensive phase of TB treatment at two months and treatment completion at six months. Samples obtained from the India cohort one year after treatment completion showed substantial overlap mediator levels between the TBDM and

TB groups, consistent with eventual resolution of TB-related inflammation in the majority of TBDM participants.

While TBDM was associated with higher levels of many plasma mediators compared to normoglycemic TB participants in the India and Brazil cohorts, lower IL-10 at baseline was the most important parameter accounting for differences between the TBDM and TB groups. IL-10 is a key anti-inflammatory mediator, inhibiting the generation of diverse pro-inflammatory cytokines in myeloid and lymphoid cells (*Robb et al., 1985*). Even in the absence of concurrent infection, hyperglycemia is associated with a systemic inflammatory state characterized by elevated ratios of diverse pro-inflammatory cytokines to IL-10 in plasma (*Butkowski and Jelinek, 2017*). Hyporesponsiveness to IL-10 signaling is an additional factor contributing to chronic low-grade inflammation in type 2 DM (*Barry et al., 2016*). A causal relationship between low IL-10 at baseline in TBDM and high levels of pro-inflammatory cannot be established with the data available from our cohort study but there was striking switch from relatively lower to higher IL-10 levels in TBDM following the initiation of ATT. While low IL-10 is commonly associated with greater susceptibility to TB (*Redford et al., 2011*), emerging evidence suggests that poorly controlled inflammation provides a host environment that supports rather than limits *Mtb* replication (*Mishra et al., 2017*; *Mishra et al., 2013*).

Among the cytokines analyzed for this study, IL-4, IL-5, IL-10, IL-13, and TGF-b participate in the resolution of infection-related pulmonary inflammation (*Barbosa et al., 2006*; *Bosurgi et al., 2017*; *Schett and Neurath, 2018*). While there was some variation in the levels of specific analytes across the cohorts, an overall trend for declining levels of pro-inflammatory mediators and increasing levels of pro-resolving mediators was observed in the TBDM and TB groups alike. The data suggest that DM comorbidity is associated with hyperinflammation prior to TB treatment, which resolves more slowly than in normoglycemic TB but is not sustained indefinitely. Plasma cytokine elevation in DM is restricted to the setting of active TB disease since Kumar et al. reported that ESAT-6 or CFP10 stimulated CD4$^+$ T cells from individuals with DM and latent TB infection have lower intracellular expression of Th1, Th2, and Th17 cytokines compared to T cells from latently infected normoglycemic individuals (*Kumar et al., 2016*).

We speculate that the excessive and prolonged inflammation in TBDM comorbidity is driven by a higher lung bacterial load stimulating innate and adaptive responses and/or a defect in counterregulation intrinsic to the diabetic host that maintains inflammatory foci despite sterilizing antimicrobial treatment. These hypotheses are not mutually exclusive. In support of the high bacterial burden hypothesis, animal studies of the TBDM interaction consistently show higher plateau levels of *Mtb* colony-forming units in the lung, along with quantitatively more pulmonary immune pathology and elevated levels of pro-inflammatory cytokines compared to normoglycemic control animals with TB (*Martens et al., 2007*; *Podell et al., 2014*). Clinical studies have variously reported higher sputum smear and culture scores and greater radiographic severity in TBDM comorbidity, which constitute crude surrogates for bacterial burden and immune pathology (*Alavi et al., 2014*; *Jiménez-Corona et al., 2013*; *Moreno-Martínez et al., 2015*; *Yoon et al., 2017*). In the current study, bacterial burden as reflected by sputum smear score was higher in TBDM than TB (*Figure 1*). In support of an intrinsic propensity for hyperinflammatory responses in TBDM, naïve T cells from chronically hyperglycemic mice were shown to have increased proliferation and production of a broad range of cytokines following antigen-receptor activation; a phenotype attributed to basal chromatin decondensation (*Martinez et al., 2014*). Furthermore, impaired wound healing is common diabetic complication that is due, in part, to impaired resolution of inflammation (*Baltzis et al., 2014*).

Our study had several limitations. While the cohorts providing clinical data and samples were recruited for prospective longitudinal investigation, the projects were not initially designed for cross-cohort comparison and radiographic scores and plasma samples from one year after TB treatment completion were not available for the Brazil cohort. The sample sizes for these two cohorts was sufficient for comparison of plasma analytes but underpowered for logistic regression. Higher median BMI and higher representation of female sex in the Brazil cohort could account at least in part for lower cytokine levels compared to the India cohort, although there was no difference in radiographic severity of disease between the two cohorts overall and more cavitary TB in the Brazil cohort (*Table 4* and *Table 5*). Finally, the normoglycemic TB arm of the EDOTS study featured high prevalence of undernutrition that is an independent risk factor for adverse TB outcomes (*Sinha et al., 2019*). This might account for the lack of differences in radiographic severity scores between the TBDM and TB groups in the India cohort.

In conclusion, we found that individuals with TBDM comorbidity not only have heightened levels of pro-inflammatory cytokines prior to ATT, but they maintain this state through the course of TB treatment. The clinical significance of excessive and prolonged inflammation in TBDM comorbidity stems from that observation that morbidity and mortality in pulmonary TB largely reflects the consequences of immune-mediated lung damage (*Ravimohan et al., 2018*). On that basis, we predict that pulmonary impairment after TB will be greater in patients with concurrent DM. We further speculate that sustained inflammation contributes to increased risk for TB-related mortality that is associated with DM (*Reed et al., 2013*). Matrix metalloproteinases (MMPs) are implicated in immune-mediated lung matrix destruction in TB (*Ong et al., 2014*). The anti-diabetic drug metformin was reported to exert a host-protective anti-inflammatory effect, including reduced MMP expression in normoglycemic mice infected with *Mtb* by aerosol, and metformin is associated with reduced risk of cavitary TB and mortality in people with concurrent DM by mechanisms that appear to be independent of any effect on glycemic control (*Degner et al., 2018*; *Singhal et al., 2014*). We previously reported that TBDM comorbidity is associated with higher plasma MMP levels compared to TB without DM, and that TBDM patients using metformin had significantly lower MMP levels than those on anti-DM treatment regimens without metformin (*Kumar et al., 2018*). Together, these observations suggest that ant-inflammatory host-directed therapies, including metformin, may be particularly beneficial in the large and growing population of TB patients with concurrent DM.

## Materials and methods

**Key resources table**

| Reagent type (species) or resource | Designation | Source or reference | Identifiers | Additional information |
|---|---|---|---|---|
| Commercial assay or kit | Multiplex ELISA, Bio-Plex Pro Human Cytokine 17-plex Assay | Bio-Rad | #m5000031yv | |

### Ethics statement

All research presented here was conducted according to the principles expressed in the Declaration of Helsinki. The India cohort study was approved by the Ethics Committee of the Prof. M. Viswanathan Diabetes Research Centre (ECR/51/INST/TN/2013/MVDRC/01). The Brazil cohort study was approved by the Ethics Committee of the Maternidade Climério de Oliveira, Federal University of Bahia (CAAE: 0115.0.054.000–09). Written informed consent was obtained from all participants at both sites.

### Indian study population

Adults (age 25–60 years) with newly diagnosed pulmonary TB disease were screened and enrolled in the ongoing EDOTS study currently underway in Chennai, India (*Kornfeld et al., 2016*). Participants with concurrent TB and DM (TBDM group) who reached the final study visit (month-18) were selected sequentially for the present investigation of plasma samples and clinical data. The comparison group of normoglycemic EDOTS participants (TB group) was matched for age and sex as closely as possible with the TBDM group. Exclusion criteria were: treatment for any prior episode of TB disease, more than 7 days treatment for the current TB episode, more than seven doses of a fluoroquinolone within 30 days, multi-drug resistant *Mtb* isolated at baseline pregnant or nursing, HIV-seropositive, or taking immunosuppressive drugs. The diagnosis TB disease was established by positive sputum culture on solid media and compatible chest x-ray at baseline. Participants were classified as having DM by HbA1c $\geq$ 6.5% for (for those with no known prior history of DM) plasma glucose $\geq$200 mg/dL at 2 hr after 75 gm glucose challenge (*American Diabetes Association, 2018*). Participants were classified as normoglycemic by plasma glucose <140 mg/dL at 2 hr after 75 gm glucose challenge according to World Health Organization (WHO) criteria. Self-reported cigarette and alcohol use was categorized as current, past, or never. A total of 43 TBDM and 44 TB participants were included in this study. Treatment for drug-sensitive TB disease was provided by government clinics and staff of the Revised National Tuberculosis Control Program in Chennai.

## Brazilian study population

Samples and clinical information used in the present study were selected from an ongoing project performed at the Instituto Brasileiro para Investigação da Tuberculose (IBIT, Brazilian Institute for TB Investigation), Salvador, Bahia, Northeast Brazil. Samples were collected between June 2015 and January 2018. For the current investigation, data from 26 TB patients with type-2 DM (HbA1c $\geq$ 6.5%) from whom cryopreserved plasma samples were available for the immunoassays at month 0 (pre-ATT), month 2 and month 6 of therapy were selected. Additional TB patients without DM (n = 25), matched by age, sex, BMI, smoking history and alcohol use were selected. Diagnosis of TB at IBIT follows the guidelines of the Brazilian Society of Pulmonology and Tisiology (*Conde et al., 2009*), which is similar to WHO recommendations. For the present study, TB diagnosis was performed at IBIT's microbiology referral laboratory as the following: three sputum smears were examined by fluorescence microscopy, processed by the modified Petroff's method and cultured on Lowenstein-Jensen medium. All patients were $\geq$18 years old, BCG-vaccinated, HIV-unexposed (confirmed with negative serology), had no prior diagnosis of TB, were diagnosed with DM at the time of study enrollment, and were not taking DM drugs. Exclusion criteria were age <18 years, previous diagnosis/treatment of TB or DM, self-reported pregnancy, and presence of psychiatric disease that might hamper proper application of the clinical questionnaire. Alcohol use was assessed by CAGE questionnaire, with alcoholism classified by positive responses to two or more questions (*Ewing, 1984*).

## Measurement of plasma mediators

Plasma samples from both cohorts were purified and stored frozen at $-80°$C prior to batch-wise Luminex assays (Bio-Rad, Hercules, CA). The parameters analyzed were interferon-$\gamma$ (IFN-g), tumor necrosis factor-$\alpha$ (TNF-a), interleukin (IL) 1-$\beta$ (IL-1b), IL-2, IL-4, IL-5, IL-6, IL-7, IL-8,IL-10, IL-12p70, IL-13, IL-17A), granulocyte colony-stimulating factor (G-CSF) and granulocyte macrophage colony-stimulating factor (GM-CSF). Plasma levels of transforming growth factor-$\beta$1 (TGF-b; R and D Systems) and IL-17F (BioLegend, San Diego, CA) were measured by ELISA.

## Statistical analysis

The median values with interquartile ranges (IQR) were used as measures of central tendency. Fisher's exact test was used to compare frequencies between the study groups. Continuous variables were compared between the study groups using the Mann-Whitney *U* test (2-group comparisons), or the Kruskall-Wallis test with Dunn's multiple comparisons ad hoc test (between three or more groups). Hierarchical cluster analyzes were performed using the Ward's method with bootstrap (100X) of z-score normalized values of soluble biomarkers. Dendograms represent Euclidean distance. A principal component analysis model using data on all the soluble biomarkers was performed to compare and visualize the grouping between TBDM and TB. Fold Change Analysis of TBDM patients versus TB patients of log2-transformed and t-test comparisons were performed with the Benjamini–Hochberg false discovery rate (FDR) adjustment for multiple testing set at 1%. Decision trees were employed to identify a minimal set of targets allowing separation between the TBDM from TB. This method analyzes all the phenotypic attributes and selects the most relevant attributes that allow group classification (*Fukutani et al., 2017*). As input for tree construction, we used data on all the markers. The J48 algorithm implemented in the WEKA program (Waikato Environment for Knowledge Analysis, version 3.6.11, University of Waikato, New Zealand). To estimate the classification accuracy of the decision tree models, we employed a 10-fold cross validation methodology. The sensibility and specificity were measured from the confusion matrix, the receiver operating characteristic curve (ROC) and viewed in the scatter plot with the decision tree cutoffs. Correlations were examined using the Spearman test. P value below 0·05 was considered statistically significant after adjustments for multiple comparisons (FDR 1%). The statistical analyses were performed using GraphPad Prism 7.0 (GraphPad Software, La Jolla, CA, USA), JMP 13.0 (SAS, Cary, NC, USA), and R 3.5.1 (R Foundation, Vienna, Austria) programs.

## Acknowledgements

The authors thank the Greater Chennai Corporation, Dr. Senthil Kumar (City Health Officer), and Dr. Lavanya J (RNTCP Program Officer) for permission to conduct the study in the TB Units of Chennai. We also thank the health care workers at IBIT, Ms. Vanessa Nascimento and sDr. Nelia Araujo, and Betania Nogueira for critical help in conducting the clinical study as well as Ms. Alice Andrade, Ms. Jessica Rebolças and Mrs. Hayna Malta, from Instituto Gonçalo Moniz (FIOCRUZ), for logistics support with the laboratory work.

## Additional information

### Funding

| Funder | Grant reference number | Author |
|---|---|---|
| CRDF Global | USB1-31149-XX-13 | Hardy Kornfeld<br>Vijay Viswanathan |
| National Institutes of Health | U01AI115940 | Bruno B Andrade |
| Conselho Nacional de Desenvolvimento Científico e Tecnológico | | Kiyoshi F Fukutani<br>Thabata Alves |
| Fundação de Amparo à Pesquisa do Estado da Bahia | | Paulo S Silveira-Mattos |
| Coordenação de Aperfeiçoamento de Pessoal de Nível Superior | | Kiyoshi F Fukutani<br>Bruno B Andrade |

The funders had no role in study design, data collection and interpretation, or the decision to submit the work for publication.

### Author contributions

Nathella Pavan Kumar, Conceptualization, Investigation, Methodology, Writing—original draft, Writing—review and editing; Kiyoshi F Fukutani, Conceptualization, Data curation, Software, Formal analysis, Validation, Visualization, Writing—original draft; Basavaradhya S Shruthi, Data curation, Investigation, Writing—original draft, Project administration; Thabata Alves, Data curation, Formal analysis; Paulo S Silveira-Mattos, Data curation, Software, Formal analysis, Methodology; Michael S Rocha, Investigation, Methodology; Kim West, Resources, Data curation, Project administration; Mohan Natarajan, Data curation; Vijay Viswanathan, Conceptualization, Resources, Supervision, Funding acquisition, Project administration; Subash Babu, Conceptualization, Supervision, Writing—original draft, Project administration, Writing—review and editing; Bruno B Andrade, Conceptualization, Resources, Software, Formal analysis, Supervision, Visualization, Writing—original draft, Project administration, Writing—review and editing; Hardy Kornfeld, Conceptualization, Supervision, Funding acquisition, Investigation, Methodology, Writing—original draft, Project administration, Writing—review and editing

### Author ORCIDs

Kim West (iD) http://orcid.org/0000-0002-5744-0280
Bruno B Andrade (iD) https://orcid.org/0000-0001-6833-3811
Hardy Kornfeld (iD) https://orcid.org/0000-0002-8970-7306

### Ethics

Human subjects: All research presented here was conducted according to the principles expressed in the Declaration of Helsinki. The Indian cohort study was approved by the Ethics Committee of the Prof. M. Viswanathan Diabetes Research Centre (ECR/51/INST/TN/2013/MVDRC/01). The Brazilian cohort study was approved by the Ethics Committee of the Maternidade Climério de Oliveira, Federal University of Bahia (CAAE: 0115.0.054.000-09). Written informed consent was obtained from all participants at both sites.

## Decision letter and Author response

Decision letter https://doi.org/10.7554/eLife.46477.019
Author response https://doi.org/10.7554/eLife.46477.020

## Additional files

### Supplementary files

• Source data 1. India Cohort.
DOI: https://doi.org/10.7554/eLife.46477.007

• Source data 2. Brazil Cohort.
DOI: https://doi.org/10.7554/eLife.46477.011

• Transparent reporting form
DOI: https://doi.org/10.7554/eLife.46477.017

### Data availability

All data generated or analyzed during this study are included in the manuscript and supporting files. Source data files have been provided for Figures 1 through 5.

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
