## [Decision Letter]

Thank you for submitting your article "Persistent inflammation during anti-tuberculosis treatment with diabetes comorbidity" for consideration by *eLife*. Your article has been reviewed by two peer reviewers, and the evaluation has been overseen by a Reviewing Editor and Satyajit Rath as the Senior Editor. The following individuals involved in review of your submission have agreed to reveal their identity: Robert Wallis (Reviewer #2); Blanca Restrepo (Reviewer #3).

The reviewers have discussed the reviews with one another and the Reviewing Editor has drafted this decision to help you prepare a revised submission.

Summary:

Diabetes Mellitus (DM) has emerged as a strong risk factor for developing active tuberculosis (TB), with associated predictions of poor outcomes and recovery of respiratory capacity. Kornfeld and colleagues undertake a longitudinal analysis of cytokines production in TB diseased individuals with and without DM in India and Brazil. The authors measure 17 plasma cytokines and growth factors in samples collected prospective from two observational cohorts of TB.

Key findings:

1) Prior to the initiation of TB treatment, individuals with DM has significantly higher cytokine levels for most of the pro-inflammatory markers measured in the blood compartment.

2) This systemic level of hyper-inflammation was sustained over the period of TB treatment.

3) The observation was noted in two distinct cohorts.

4) There appears to be a delay in the induction of pro-resolving cytokines (that dampen inflammation) in individuals with DM.

Conclusion:

Individuals with DM and TB have higher levels of system inflammation, at baseline and completion of TB treatment, than those with TB alone. This high level of inflammation most likely contributes to pathogenesis and disease severity, an effect that can be exploited by host directed therapies that modulate inflammation.

The following needs to be addressed:

Major comments.

1) The paper will benefit from additional information to help resolve the question as to whether persistent inflammation was due to persistent infection. This could include baseline indicators of heavy bacterial burden, or evidence of delayed culture conversion during treatment.

2) State the proportion of patients whose DM had already been diagnosed and treatment initiated prior to the TB diagnosis, and whether outcomes in these patients were different in any way.

3) The conclusion of longer persistence of inflammation is supported by the findings in the Indian cohort, but weakly corroborated by the Brazilian cohort. While pro-inflammatory cytokines do persist in the Brazilian cohort, there are only a few cytokines up-regulated at a given time point, and there is no consistency for any given cytokine that remains high throughout the treatment. Can the authors provide an explanation?

4) The different results between both cohorts with respect to TBDM vs TB may be due to "genetic, demographic, and behavioural characteristics" as the authors point out. Please expand on the major demographic differences between the TBDM vs TB group in India, such as older age and higher BMI, which are known to be associated with higher inflammation (independent of diabetes). It is unclear that the authors account for these important differences in their statistical analysis. In contrast, careful matching of TBDM vs TB was done for these factors in the Brazilian cohort. Therefore, these demographic variations between cohorts may explain the clear differences in inflammation in the Indian cohort, but weaker differences in the Brazilian cohort. Clarify the influence of diabetes versus other host factors on the observed inflammation.

5) The Results section is misleading regarding the method for DM classification at both study sites. It is only in the Materials and methods that the reader realizes that the oral glucose tolerance test was only done in those without DM history in the Indian cohort, but not the Brazilian cohort. The Results section needs to be edited accordingly.

6) Figure 3B. It would be easier to follow the change in pro- and anti-inflammatory cytokines, if listed in such an order (similar to the order reported for the Indian cohort in Figure 2B).

7) "[…] switch from relatively higher to relatively lower IL-10 levels in TBDM following the initiation of ATT". "higher" and "lower" seemed to be the opposite of what was reported.

8) Discussion. This sentence is unclear: "Further supporting the notion that DM is not invariably associated with hyper-responsiveness to Mtb, Kumar et al. reported that ESAT-6 or CFP10 stimulated CD4^+^ T cells from individuals with DM and latent TB infection have lower intracellular expression of Th1, Th2, and Th17 cytokines compared to T cells from latently infected normoglycemic individuals (Kumar et al., 2016).

---

## [Author Response]

Major comments.1) The paper will benefit from additional information to help resolve the question as to whether persistent inflammation was due to persistent infection. This could include baseline indicators of heavy bacterial burden, or evidence of delayed culture conversion during treatment.

We thank the reviewer for the suggestion. We performed comparisons, now shown in the Results (Figure 1), demonstrating that sputum smear grades were higher in TBDM patients compared to normoglycemic TB controls. In addition, we compared frequency of individuals remaining culture‐positive at month 2 of anti‐TB treatment but found no difference between TBDM and TB groups in India or in Brazil.

2) State the proportion of patients whose DM had already been diagnosed and treatment initiated prior to the TB diagnosis, and whether outcomes in these patients were different in any way.

All participants in the Brazilian cohort were newly diagnosed with DM at the time of TB screening and none were taking antidiabetic drugs. Of 43 participants in the Indian cohort, 29 were diagnosed with DM prior to incident TB while 14 were newly diagnosed with DM when screened for enrolment in the EDOTS study. Of the 29 known‐DM participants, 23 were taking some form of antidiabetic treatment at the time of enrolment, including 8 taking metformin and the remaining 15 on diabetes treatment regimens that did not contain metformin. TB treatment outcomes were similar between the known‐DM and new‐DM individuals; all competed the 6‐month anti‐TB regimen with bacteriological cure. Two of 29 known‐DM and one of 15 new‐DM participants had recurrent TB within one year after treatment completion. The revised manuscript presents a comparison of new vs known DM participant characteristics (Table 2) as well as a new figure showing that the fold‐change in cytokines of new vs known DM participants was not significantly different (Figure 2—figure supplement 1). This additional analysis does not alter our original conclusions.

3) The conclusion of longer persistence of inflammation is supported by the findings in the Indian cohort, but weakly corroborated by the Brazilian cohort. While pro-inflammatory cytokines do persist in the Brazilian cohort, there are only a few cytokines up-regulated at a given time point, and there is no consistency for any given cytokine that remains high throughout the treatment. Can the authors provide an explanation?

While there are overall quantitative differences between cytokine levels in the Indian and Brazilian cohorts, the important finding conveyed in our analysis is that in both cohorts, the TBDM and TB groups were identifiable by hierarchical clustering and principal component analysis as discrete populations based on cytokine levels at all time points during TB treatment. Virtually all clinical studies in human populations find high degrees of individual variability in the levels of specific cytokines, which reflect a very broad range of intrinsic and extrinsic factors. The important point from our data is that systemic inflammation as reflected by plasma cytokine levels is persistently higher in TBDM than in TB patients before and through the course of TB treatment.

4) The different results between both cohorts with respect to TBDM vs TB may be due to "genetic, demographic, and behavioural characteristics" as the authors point out. Please expand on the major demographic differences between the TBDM vs TB group in India, such as older age and higher BMI, which are known to be associated with higher inflammation (independent of diabetes). It is unclear that the authors account for these important differences in their statistical analysis. In contrast, careful matching of TBDM vs TB was done for these factors in the Brazilian cohort. Therefore, these demographic variations between cohorts may explain the clear differences in inflammation in the Indian cohort, but weaker differences in the Brazilian cohort. Clarify the influence of diabetes versus other host factors on the observed inflammation.

New tables (Table 4 and Table 5) in the revised manuscript provide additional details in the comparison of the cohorts from India and Brazil. The factors with statistically significant differences that might also be biologically significant in the context of TB are higher mean BMI and a greater proportion of female participants in the Brazilian cohort. While these factors might attenuate the severity of TB disease and contribute to lower cytokine levels in the Brazilian vs Indian cohorts, that would not alter our conclusion that diabetes is associated with a higher level of systemic inflammation that persists through the course of TB treatment. That this finding was made despite differences in the cohorts only strengthens that conclusion.

5) The Results section is misleading regarding the method for DM classification at both study sites. It is only in the Materials and methods that the reader realizes that the oral glucose tolerance test was only done in those without DM history in the Indian cohort, but not the Brazilian cohort. The Results section needs to be edited accordingly.

The Results section has been revised to indicate that DM classification in the Brazilian cohort was based on HbA1c only.

6) Figure 3B. It would be easier to follow the change in pro- and anti-inflammatory cytokines, if listed in such an order (similar to the order reported for the Indian cohort in Figure 2B).

The analyses were performed using unsupervised hierarchical clustering, which orders the markers based on concentration profiles in the study groups/timepoints. We understand that the reviewer’s point, but the nature of our analyses precludes changing the order.

7) "[…] switch from relatively higher to relatively lower IL-10 levels in TBDM following the initiation of ATT". "higher" and "lower" seemed to be the opposite of what was reported.

This error has been corrected in the revised manuscript.

8) Discussion. This sentence is unclear: "Further supporting the notion that DM is not invariably associated with hyper-responsiveness to Mtb, Kumar et al. reported that ESAT-6 or CFP10 stimulated CD4^+^ T cells from individuals with DM and latent TB infection have lower intracellular expression of Th1, Th2, and Th17 cytokines compared to T cells from latently infected normoglycemic individuals (Kumar et al., 2016).

Our intention was to point out that in the absence of active TB disease, individuals with DM do not have increased levels of plasma cytokines. This was included to address the question whether cytokine elevation is an inherent feature DM alone. The text has been revised to try make this point more clearly.